# In-context Autoencoder for Context Compression in a Large Language Model

**Tao Ge**[*]  **Jing Hu**[†]  **Lei Wang**[†]  **Xun Wang**  **Si-Qing Chen**  **Furu Wei**
Microsoft Corporation
{tage,v-hjing,v-leiwang7,xunwang,sqchen,fuwei}@microsoft.com

## Abstract

We propose the In-context Autoencoder (ICAE), leveraging the power of a large language model (LLM) to compress a long context into short compact memory slots that can be directly conditioned on by the LLM for various purposes. ICAE is first pretrained using both autoencoding and language modeling objectives on massive text data, enabling it to generate memory slots that accurately and comprehensively represent the original context. Then, it is fine-tuned on instruction data for producing desirable responses to various prompts. Experiments demonstrate that our lightweight ICAE, introducing about 1% additional parameters, effectively achieves $4\times$ context compression based on Llama, offering advantages in both improved latency and GPU memory cost during inference, and showing an interesting insight in memorization as well as potential for scalability. These promising results imply a novel perspective on the connection between working memory in cognitive science and representation learning in LLMs, revealing ICAE's significant implications in addressing the long context problem and suggesting further research in LLM context management. Our data, code and models are available at https://github.com/getao/icae.

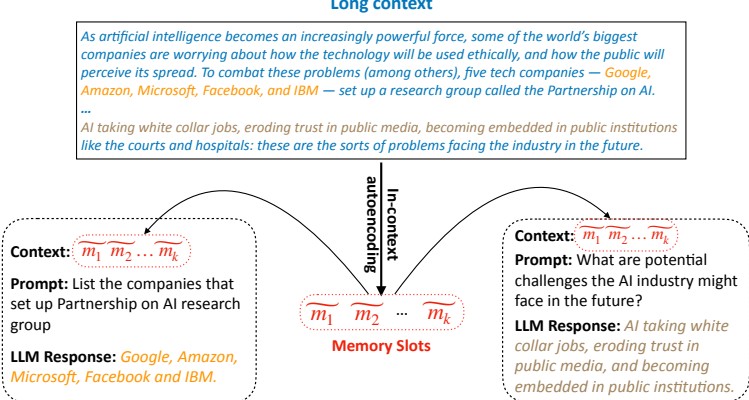

Figure 1: Compressing a long context into **a short span of memory slots**. The memory slots can be conditioned on by the target LLM on behalf of the original context to respond to various prompts.

## 1 Introduction

Long context modeling is a fundamental challenge for Transformer-based (Vaswani et al., 2017) LLMs due to their inherent self-attention mechanism. Much previous research (Child et al., 2019; Beltagy et al., 2020; Rae et al., 2019; Choromanski et al., 2020; Bulatov et al., 2022; Zheng et al., 2022; Wu et al., 2022; Bulatov et al., 2023; Ding et al., 2023) attempts to tackle the long context issue through architectural innovations of an LLM. While they approach long context with a significant reduction in computation and memory complexity, they often struggle to overcome the notable decline

---

[*]Correspondence to Tao Ge (sggetao@gmail.com)
[†]Internship at Microsoft Research

in performance on long contexts, as highlighted by Liu et al. (2023). In contrast to these efforts, we approach the long context problem from a novel angle – context compression.

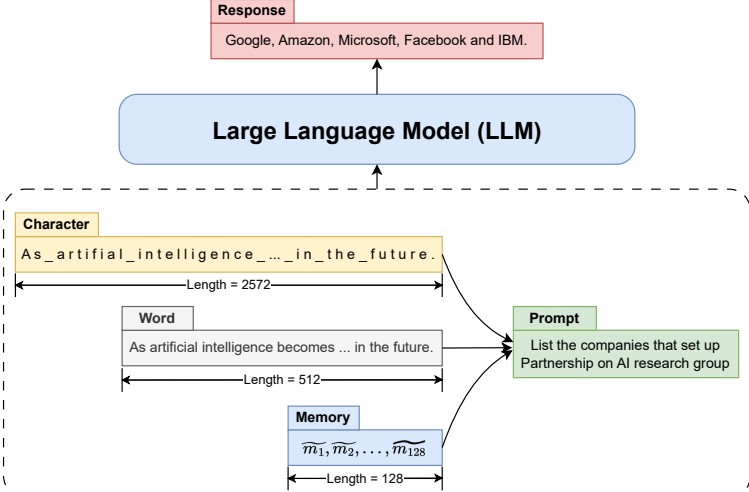

Figure 2: Various context lengths (e.g., 2572 chars, 512 words, 128 memory slots) serve the same function when conditioned on by an LLM for responding to the given prompt.

Context compression is motivated by the fact that a text can be represented in different lengths in an LLM while conveying the same information. As shown in Figure 2, if we use characters to represent the text, it will have a length of 2,572; if we represent it using (sub-)words, we only need a context length of 512 without affecting the response accuracy. So, is there a more compact representation allowing us to achieve the same goal with a shorter context?

We explore this problem and propose the ICAE which leverages the power of an LLM to achieve high compression of contexts. The ICAE consists of 2 modules: a learnable encoder adapted from the LLM with LoRA (Hu et al., 2021) for encoding a long context into a small number of memory slots, and a fixed decoder, which is the LLM itself where the memory slots representing the original context are conditioned on to interact with prompts to accomplish various goals, as illustrated in Figure 1.

We first **pretrain** the ICAE using both autoencoding (AE) and language modeling (LM) objectives so that it can learn to generate memory slots from which the decoder (i.e., the LLM) can recover the original context or perform continuation. The pretraining with massive text data enables the ICAE to be well generalized, allowing the resulting memory slots to represent the original context more accurately and comprehensively. Then, we **fine-tune** the pretrained ICAE on instruction data for practical scenarios by enhancing its generated memory slots' interaction with various prompts. We show the ICAE (based on Llama) learned with our pretraining and fine-tuning method can effectively produce memory slots with $4\times$ context compression. We highlight our contributions as follows:

- We propose In-context Autoencoder (ICAE) – a novel approach to context compression by leveraging the power of an LLM. The ICAE either enables an LLM to express more information with the same context length or allows it to represent the same content with a shorter context, thereby enhancing the model's ability to handle long contexts with improved latency and memory cost during inference. Its promising results and its scalability may suggest further research efforts in context management for an LLM, which is orthogonal to other long context modeling studies and can be combined with them to further improve the handling of long contexts in an LLM.

- In addition to context compression, ICAE provides an access to probe how an LLM performs memorization. We observe that extensive self-supervised learning (e.g., autoencoding) in the pretraining phase is very helpful to enhance the ICAE's capability to encode the original context into compressed memory slots. This pretraining process may share some analogies with humans enhancing their memory capacity through extensive memory training, which improves the brain's memory encoding capabilities (Ericsson et al., 1980; Engle et al., 1999; Maguire et al., 2003). We also show that an LLM's memorization pattern is highly similar to humans (see Table 2 and Table 3). All these results imply a novel perspective on the connection between working memory in cognitive science (Baddeley, 1992) and representation learning in LLMs (i.e., context window).

## 2 IN-CONTEXT AUTOENCODER

### 2.1 MODEL ARCHITECTURE

Like a typical autoencoder (Kramer, 1991), ICAE consists of an encoder and a decoder. Similar to the design of Gisting (Mu et al., 2023) and AutoCompressor (Chevalier et al., 2023), the ICAE performs both the encoding and decoding processes in an in-context manner, as illustrated in Figure 3.

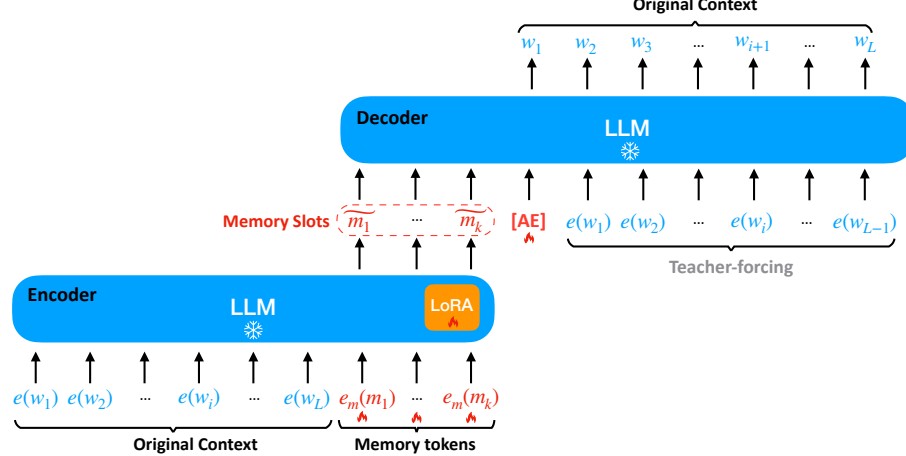

Figure 3: The encoder of the ICAE is a LoRA-adapted LLM, which is used for encoding the original context $c = (w_1, w_2, \ldots, w_L)$ into a few memory slots $(\widetilde{m_1}, \ldots, \widetilde{m_k})$. The decoder of the ICAE is the target LLM itself that can condition on the memory slots produced by the encoder for various purposes (e.g., the autoencoding task as in this figure). $e(\cdot)$ denotes the word embedding lookup in the target LLM and $e_m(\cdot)$ denotes the learnable embedding lookup of memory tokens that are used for producing memory slots. "[AE]" is a special token to indicate the autoencoding pretraining task.

Given the intuition, we propose to use a LoRA-adapted LLM as the encoder of the ICAE, as illustrated in Figure 3. When encoding a context $c = (w_1, \ldots, w_L)$ with the length $L$, we first append $k$ ($k << L$) memory tokens $(m_1, \ldots, m_k)$ to the context $c$ to obtain their outputs $(\widetilde{m_1}, \ldots, \widetilde{m_k})$ as the memory slots for the context $c$. Therefore, the ICAE encoder is very lightweight – it only adds a LoRA adapter and an embedding lookup for memory tokens compared with the target LLM.

As introduced above, we expect the memory slots $(\widetilde{m_1}, \ldots, \widetilde{m_k})$ to be conditioned on by the target LLM on behalf of the original context $c$. Therefore, we use the untouched target LLM as the decoder of the ICAE to ensure the compatibility of memory slots within the target LLM.

### 2.2 PRETRAINING

#### 2.2.1 AUTOENCODING

Like a typical autoencoder, one of the ICAE's pretraining objectives is to restore the original input text $c$ of the length $L$ from its produced memory slots $(\widetilde{m_1}, \ldots, \widetilde{m_k})$ of the length $k$:

$$\mathcal{L}_{\text{AE}} = \max_{\widetilde{m_1}, \ldots, \widetilde{m_k}} P(c|\widetilde{m_1}, \ldots, \widetilde{m_k}; \Theta_{LLM}) = \max_{\Theta_{LoRA}, e_m} P(c|m_1 \ldots m_k; \Theta_{LLM}, \Theta_{LoRA}, e_m)$$

To indicate the autoencoding task, we append a special token "[AE]" to $(\widetilde{m_1}, \ldots, \widetilde{m_k})$ in the decoder, as Figure 3 shows. As this pretraining objective does not need any extra annotation, we can use massive text data to train the In-context Autoencoder.

#### 2.2.2 TEXT CONTINUATION

While autoencoding pretraining offers a straightforward learning objective to encode a context, its inherent simplicity and exclusive focus on the single objective may lead to suboptimal generalization. To address this issue, we incorporate an additional objective during the pretraining phase: text continuation, as illustrated in Figure 7 in Appendix A. This self-supervised task is widely acknowledged to facilitate the learning of more generalizable representations in language models:

$$\mathcal{L}_{\text{LM}} = \max_{\widetilde{m_1}, \ldots, \widetilde{m_k}} P(o|\widetilde{m_1}, \ldots, \widetilde{m_k}; \Theta_{LLM}) = \max_{\Theta_{LoRA}, e_m} P(o|m_1 \ldots m_k; \Theta_{LLM}, \Theta_{LoRA}, e_m)$$

where $\boldsymbol{o} = (w_{L+1}, \ldots, w_{L+N})$ denotes the continuation of context $\boldsymbol{c}$. This objective helps improve generalization and circumvent excessive reliance on, and overfitting to, the autoencoding task.

## 2.3 INSTRUCTION FINE-TUNING

After pretraining, the memory slots produced by the pretrained ICAE are expected to represent the original context. However, for LLMs, the purpose of providing a context extends beyond rote memorization or continuation; instead, the more common use scenario is using the provided context as a basis for accurately and appropriately responding to various prompts, ultimately accomplishing the tasks we want it to perform (Wei et al., 2021; Ouyang et al., 2022).

To enhance the interaction of memory slots produced by the ICAE with diverse prompts, we further fine-tune the ICAE with the PwC dataset (**P**rompt-**w**ith-**C**ontext), a dataset[1] introduced in this paper consisting of thousands of (context, prompt, response) samples (as shown in Figure 1).

Formally, the ICAE is fine-tuned for learning to encode the context into the memory slots based on which the decoder (i.e., the target LLM) can produce a desirable response $r_1 \ldots r_n$ according to a given prompt $p_1 \ldots p_m$, as shown in Figure 8 in Appendix A:

$$\mathcal{L}_{\text{FT}} = \max_{\widetilde{m_1} \ldots \widetilde{m_k}} P(r_1 \ldots r_n | \widetilde{m_1} \ldots \widetilde{m_k}, p_1 \ldots p_m; \Theta_{LLM})$$
$$= \max_{\Theta_{LoRA}, e_m} P(r_1 \ldots r_n | m_1 \ldots m_k, p_1 \ldots p_m; \Theta_{LLM}, \Theta_{LoRA}, e_m)$$

## 3 EXPERIMENTS

### 3.1 EXPERIMENTAL SETTING

**Data**   We pretrain the ICAE with the Pile (Gao et al., 2020). For instruction fine-tuning, we use the PwC dataset, as introduced in Section 2.3, which contains 240k (context, prompt, response) samples for training and 18k samples for testing. The context length distribution of test samples is shown in Figure 10. By default, the maximal token length (excluding memory slots) we set during training is 512 in both the ICAE's encoder and decoder in our experiments.

**Model Configuration**   We use the LlaMa (Touvron et al., 2023a;b) as the target LLM to test the ICAE's performance in context compression. For the encoder of the ICAE, LoRA is applied to the query and value projections of the LLM's multi-head attention. In our default setting, the memory slot length $k$ is set to 128, and the LoRA rank $r$ is set to 128 unless otherwise specified. The resulting ICAE only adds about 1% learnable parameters on top of the target LLM.

### 3.2 RESULTS

#### 3.2.1 PRETRAINED ICAE

We first evaluate the autoencoding performance of the pretrained ICAE (without instruction fine-tuning) using the following three metrics to understand how well it restores the original context from its produced memory slots: BLEU (Papineni et al., 2002), Exact-Match (EM)[2] and cross entropy loss.

Figure 4 presents the autoencoding results of the ICAE based on the Llama-7b. The ICAE demonstrates a very low overall loss, below 0.05, indicating that the produced memory slots retain almost all the information of the original context. When the context length is within 300, the ICAE can almost perfectly reconstruct the original context, achieving nearly 100% BLEU and EM scores. As the context length increases beyond 400, both BLEU and EM scores start to decline, indicating insufficient capacity of the 128-length memory slots. However, even at a context length of 500, the median BLEU remains over 0.98, and the median EM approaches 0.6 (e.g., perfectly reconstructing about the first 300 words of a 512-token context), showing remarkable performance of ICAE.

We then analyze the effect of the memory size $k$ on the result. According to Figure 5, as the memory slot length $k$ decreases, the ICAE's ability to memorize longer samples significantly deteriorates.

---

[1] Despite some (prompt, response) datasets such as Self-Instruct (Wang et al., 2022), most of their samples either have no context or very short contexts, which are not suitable for evaluation in our setting. Therefore, we establish the PwC dataset with the help of the GPT-4 (OpenAI, 2023). We include the details in Appendix C.

[2] EM denotes the proportion of the exact matching prefix length to the total length. For a context of 512 tokens, if its first 256 tokens are perfectly restored but its 257th token is not, the EM score is $256/512 = 0.5$.

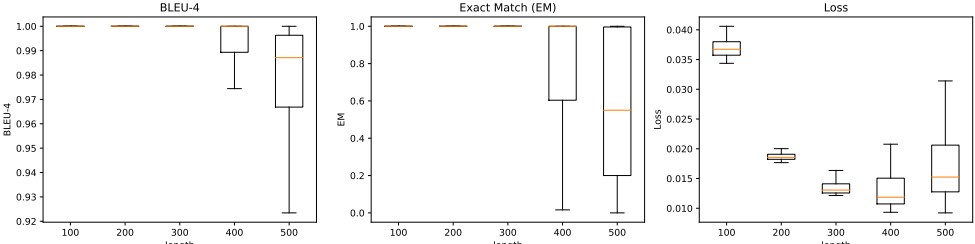

Figure 4: Autoencoding results of the ICAE based on the Llama-7b with memory length $k = 128$. The horizontal axis represents the original context length of test examples. For example, the horizontal axis value of 100 refers to the test examples with context lengths ranging from 95 to 105.

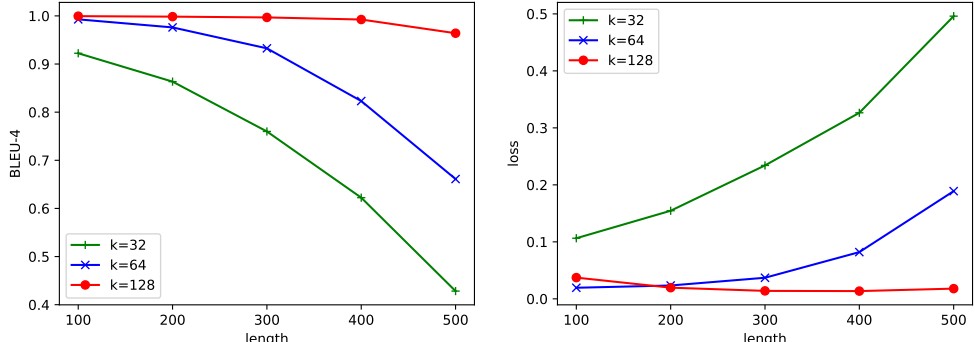

Figure 5: BLEU and loss at different memory slot lengths $k$.

Compared to $k = 128$ where the BLEU score can still reach over 95% at a context length of 500, the BLEU scores become much less satisfactory for $k$ values of 64 and 32, indicating an inability to losslessly retain the original context. This observation is also evident from the loss curve, suggesting that achieving over $4\times$ compression is rather challenging.

Table 1: Text continuation evaluation for the pretrained ICAE. Similar to the autoencoding evaluation, a higher compression ratio tends to result in more pronounced losses in language modeling.

| Context length | Text Continuation | | |
| --- | --- | --- | --- |
| | PPL (w/ original context) | PPL (w/ 128 memory slots) | $\Delta$ |
| 128→128 (1×) | 9.99 | 10.15 | +0.16 |
| 256→128 (2×) | 9.45 | 9.77 | +0.32 |
| 512→128 (4×) | 9.01 | 9.50 | +0.49 |

Similarly, the text continuation evaluation presented in Table 1 also illustrates that a higher compression ratio tends to result in more pronounced losses in language modeling.

Table 2 presents 1 specific example of the ICAE performing text restoration, demonstrating an interesting behavior: "*large pretrained language model*" is restored as "*large pretrained model*" and "*The results prove*" is restored as "*The experimental evidence proves*". These restoration errors resemble mistakes humans would make when memorizing the same text. This suggests that, like humans, the model selectively emphasizes or neglects certain parts of the information during the memorization based on its own understanding. It is also consistent with Peng et al. (2023): the stronger the LLM, the fewer it needs to memorize, and thus the smaller the memorization effort. This is similar to human learning: knowledgeable individuals tend to learn more effortlessly, while those with limited knowledge often rely on rote memorization to acquire new information.

To further look into the memorization insight, we test restoration performance for different types of 512-token texts with 128 memory slots produced by ICAE to investigate whether its memorization capability is consistent across different content types. According to Table 3, in contrast to compressing normal texts which can be well restored, compressing and restoring less common texts (i.e., random texts) becomes very challenging, reflected by much worse loss and BLEU scores. All these results strongly support our intuition that an LLM's memorization pattern is highly similar to humans.

Table 2: 1 example showing how the pretrained ICAE ($k = 128$) restores the original context.

| Origin Context | Restoration |
|---|---|
| Large pretrained ==language== models have shown surprising In-Context Learning (ICL) ability. With a few demonstration input-label pairs, they can predict the label for an unseen input without additional parameter updates. Despite the great success in performance, the working mechanism of ICL still remains an open problem. In order to better understand how ICL works, this paper explains language models as meta-optimizers and understands ICL as a kind of implicit finetuning. Theoretically, we figure out that the Transformer attention has a dual form of gradient descent based optimization. On top of it, we understand ICL as follows: GPT first produces metagradients according to the demonstration examples, and then these meta-gradients are applied to the original GPT to build an ICL model. Experimentally, we comprehensively compare the behavior of ICL and explicit finetuning based on real tasks to provide empirical evidence that supports our ==understanding==. The ==results prove== that ICL behaves ==similarly to explicit finetuning== at the ==prediction== level, the representation level, and the attention behavior level. Further, inspired by our understanding of meta-optimization, we design a momentum-based attention by analogy with the ==momentum-based gradient descent== algorithm. Its consistently better performance ==over== vanilla attention supports ==our understanding== again from another aspect, and more importantly, it shows the potential to ==utilize== our understanding for future ==model designing==. | Large pretrained models have shown surprising In-Context Learning (ICL) ability. With a few demonstration input-label pairs, they can predict the label for an unseen input without additional parameter updates. Despite the great success in performance, the working mechanism of ICL still remains an open problem. In order to better understand how ICL works, this paper explains ==how== language models as meta-optimizers and understands ICL as a kind of implicit finetuning. Theoretically, we figure out that the Transformer attention has a dual form of gradient descent based ==on== optimization. On top of it, we understand ICL as follows: GPT first produces metagradients according to the demonstration examples, and then these meta-gradients are applied to the original GPT to build an ICL model. Experimentally, we comprehensively compare the behavior of ICL and explicit finetuning based on real tasks to provide empirical evidence that supports our ==findings==. The ==experimental evidence proves== that ICL behaves ==like us to the same extent.== ==Prediction== at the explicit ==finetuning== level, the representation level, and the attention behavior level. Further, inspired by our understanding of meta-optimization, we design a momentum-based attention by analogy with the ==gradient descent-based momentum gradient== algorithm. Its consistently better performance ==against== vanilla attention supports ==us== again from another aspect, and more importantly, it shows the potential to ==use== our understanding for future ==modeling tasks==. |

Table 3: Restoration performance for different types of 512-token content with 128 memory slots. Patterned random text is obtained by adding 1 to each token_id in a normal text.

| Content type | Loss | BLEU |
|---|---|---|
| Normal text | 0.01 | 99.3 |
| Patterned random text | 1.63 | 3.5 |
| Completely random text | 4.55 | 0.2 |

Based on this intuition, it is very likely that a more powerful LLM may support a higher compression ratio without significant forgetting. We will discuss it in Section 3.3.1.

### 3.2.2 FINE-TUNED ICAE

In order to evaluate the fine-tuned ICAE's performance, we evaluate on the PwC test set. We use the GPT-4 to compare the outputs of the two systems to determine which one performs better or if they are on par with each other, following Mu et al. (2023). Table 4 shows the comparison of results of the LLMs conditioned on memory slots and original contexts. For Llama-7b (fine-tuned ICAE), we compare with Alpaca and StableLM-tuned-alpha-7b since there is no official instruction-tuned Llama-1 model. The Llama-7b (ICAE) conditioned on 128 memory slots largely outperforms both Alpaca and StableLM which can access original contexts ($\sim$512 tokens), with a win rate of 56.7% and 74.1% respectively and a win+tie rate of 73%$\sim$81%. However, when compared to the GPT-4 (we regard it as the gold standard), there is still a significant gap, with around 70% of the cases underperforming the GPT-4's results, and a win+tie ratio of about only 30%.

When we switch the base model to Llama-2-chat, we observe ICAE's performance becomes much better than its counterpart based on Llama-1: when $k = 128$, its win+tie rate can reach around 75% againt the GPT-4 although it still lags behind its counterpart conditioning on the original context as the compression is lossy. As $k$ increases, the win+tie rate further improves while the compression rate decreases. We perform the same comparative studies on Llama-2-13b-chat and observe better results of ICAE, supporting our assumption in Section 3.2.1 that the ICAE can benefit more on larger LLMs.

We investigate the impact of memory length on results. Table 5 shows pairwise comparisons between ICAE models with varying memory slot lengths. A higher compression ratio makes it harder to ensure response quality, but a larger ratio doesn't always lead to worse performance. Table 5 highlights that a pretrained ICAE with $8\times$ compression ($k$=64) can match a non-pretrained ICAE with $4\times$ compression ($k$=128). Under the same ratio, the pretrained ICAE performs much better than its

Table 4: Memory slots *VS* Original contexts ($\sim$512 tokens) on the PwC test set

| System 1 (k memory slots) | System 2 (original context) | Judgement (%) | | | |
|---|---|---|---|---|---|
| | | win | lose | tie | on par (win+tie) |
| Llama-7b (ICAE, $k$=128) | Alpaca | 56.7 | 26.9 | 16.4 | 73.1 |
| | StableLM-7b | 74.1 | 18.8 | 7.2 | 81.3 |
| | GPT-4 (gold) | 3.4 | 69.4 | 27.2 | 30.6 |
| Llama-2-7b-chat (ICAE, $k$=64) | Llama-2-7b-chat | 13.6 | 51.6 | 34.8 | 48.4 |
| | GPT-4 (gold) | 1.9 | 44.7 | 53.4 | 55.3 |
| Llama-2-7b-chat (ICAE, $k$=128) | Llama-2-7b-chat | 19.6 | 45.4 | 35.0 | 54.6 |
| | GPT-4 (gold) | 2.8 | 25.8 | 71.4 | 74.2 |
| Llama-2-7b-chat (ICAE, $k$=256) | Llama-2-7b-chat | 22.0 | 22.2 | 55.8 | 77.8 |
| | GPT-4 (gold) | 3.8 | 20.5 | 75.7 | 79.5 |
| Llama-2-13b-chat (ICAE, $k$=256) | Llama-2-13b-chat | 21.9 | 20.8 | 57.3 | 79.2 |
| | GPT-4 (gold) | 4.0 | 19.2 | 76.8 | 80.8 |

Table 5: ICAE with different memory slot lengths and different pretraining setups. The last row is the comparison between 128-length ICAE's memory and 128-token summary produced by the GPT-4.

| ICAE (Llama-2-7b-chat) | Judgement | | | |
|---|---|---|---|---|
| | win (%) | lose (%) | tie (%) | win/lose |
| $k = 128$ (pretrained) **VS** $k = 64$ (pretrained) | 57.6 | 19.5 | 22.9 | 3.0 |
| $k = 64$ (pretrained) **VS** $k = 32$ (pretrained) | 44.7 | 21.8 | 33.5 | 2.1 |
| $k = 64$ (pretrained) **VS** $k = 128$ (no pretraining) | 33.1 | 28.0 | 38.9 | 1.2 |
| $k = 128$ (pretrained) **VS** $k = 128$ (no pretraining) | 60.4 | 9.5 | 30.1 | 6.4 |
| $k = 128$ (pretrained) **VS** $k = 128$ (pretrained only with AE) | 36.4 | 28.5 | 35.1 | 1.3 |
| $k = 128$ (pretrained) **VS** $k = 128$ (pretrained only with LM) | 35.1 | 24.9 | 40.0 | 1.4 |
| $k = 128$ (pretrained) **VS** 128-token summary (by GPT-4) | 34.1 | 17.6 | 48.3 | 1.9 |

non-pretrained counterpart, emphasizing the importance of pretraining. By comparing the outputs generated via the pretrained and non-pretrained ICAE, we find the pretrained ICAE suffers less from hallucination than the non-pretrained counterpart (see the examples in Table 9 in Appendix D). We assume the pretraining of ICAE improves the LLM's working memory as it shares some analogies with humans enhancing their memory capacity via extensive memory training which improves the brain's memory encoding capabilities. We also examine pretraining objectives and find combining[3] AE and LM yields better results than using AE or LM individually (the 4th row in Table 5).

The last row of Table 5 compares ICAE's 128-length memory slots with a summary[4] within 128 tokens ($\sim$100 words). Memory slots significantly outperform summaries under the same context length, with $\sim$2$\times$ win/lose ratio, proving to be more compact and informative than natural language.

### 3.3 ANALYSIS

#### 3.3.1 SCALABILITY

As discussed above, ICAE should achieve better compression performance with a more powerful target LLM. To verify this assumption, we compare the ICAE's performance on three target LLMs: Llama-7b, Llama-2-7b and Llama-2-13b in Table 6, which align well with our expectations – more powerful target LLMs can achieve better context compression ratios.

#### 3.3.2 LATENCY

We conducted an empirical test to evaluate the impact of ICAE's $4\times$ context compression on inference efficiency. For this efficiency test, we fix the context (i.e., input) length to either 512 or 2048 and the generation length to 128. Table 7 shows that context compression by ICAE is helpful to improve LLM (i.e., Llama-7b) inference efficiency, achieving over $2\times$ speedup. Its acceleration becomes

---

[3] $\mathcal{L}_{\text{pretrain}} = \lambda \mathcal{L}_{\text{AE}} + (1 - \lambda)\mathcal{L}_{\text{LM}}$. We find $\lambda = 0.4 \sim 0.6$ leads to the best result.

[4] Produced by the GPT-4. The specific prompt text is presented in Appendix D.

Table 6: The results of pretrained ICAE (512→128) based on different target LLMs

| Target LLM | AE | | Text Continuation | | |
|---|---|---|---|---|---|
| | BLEU(%) | Loss | PPL (original context) | PPL (memory slot) | Δ |
| Llama-7b | 99.1 | 0.017 | 9.01 | 9.50 | +0.49 |
| Llama-2-7b | 99.5 | 0.009 | 8.81 | 9.18 | +0.37 |
| Llama-2-13b | 99.8 | 0.004 | 8.15 | 8.45 | +0.30 |

Table 7: Latency comparison of LLM (generation) and LLM+ICAE (compression then generation)

| Input (Batch×Length) | Method | Compression Time (Cachable) | Decoding Time | Total Time |
|---|---|---|---|---|
| 8*2048 | LLM | - | 24.0 | 24.0 |
| | LLM+ICAE | 3.4 | 3.9 | 7.3 (3.3×) |
| 8*512 | LLM | - | 9.3 | 9.3 |
| | LLM+ICAE | 0.6 | 3.7 | 4.3 (2.2×) |
| 32*512 | LLM | - | 24.3 | 24.3 |
| | LLM+ICAE | 2.6 | 4.2 | 6.8 (3.6×) |

even more significant – around 3.5× – in compute-intensive scenarios (e.g., 8×2048 and 32×512). Given that the compressed memory slots can be cached in advance (for frequently used texts like textbooks, government reports or articles of law), ICAE may introduce over 7× inference speedup in these cases. Details of the profiling are presented in Appendix B.

### 3.3.3 MULTIPLE SPANS OF MEMORY SLOTS

Thus far, we have mainly discussed a single span of memory slots. In this section, we shall discuss multiple spans of memory slots. As illustrated in Figure 6(Left), we can segment a long context into $N$ chunks, compress them individually, and then concatenate them to represent the original long context. However, this did not work initially, because the model had never seen multiple span concatenation patterns during training. Fortunately, we can incorporate a small number of multiple span concatenation samples during training, enabling the model to work with concatenated spans of memory slots, as OpenAI's work (Bavarian et al., 2022) on introducing the "fill in the middle" ability for the GPT. The results in Figure 6(Right) indicate that, using an equivalent length context, ICAE's memory achieves better performance – because memory can represent 4× the original context length.

The ability of ICAE demonstrates great promise to handle long contexts, as it can save a significant amount of GPU memory when addressing long contexts without touching the existing LLM. As illustrated in Figure 6(Right), 2048-length memory slots can perform on par with 4096-token contexts. This means that conditioning on 2048 memory slots instead of the original 4096 context tokens can save about 20GB of GPU memory[5] with minimal quality degradation.

## 4 RELATED WORK

Prompt compression and context distillation (Askell et al., 2021; Snell et al., 2022) are closely related areas to this work: Wingate et al. (2022) proposed a method to learn compact soft prompts to simulate the original natural language prompt by optimizing the KL divergence. However, this approach has a very high computational cost, as it requires performing back-propagation for each new incoming prompt to learn and obtain the compressed prompt, which severely limits its application. Qin & Van Durme (2023) propose Neural Agglomerative Embeddings named NUGGET, which encodes language into a compact representation for an encoder-decoder model.

The most closely related studies to our research are GIST (Mu et al., 2023) and AutoCompressors (Chevalier et al., 2023). GIST achieves prompt compression by fine-tuning an LLM in a similar way to ours. The resulting model can produce gist tokens as the compression of a prompt, which are similar to our memory slots. Nonetheless, this approach is limited to compressing short prompts[6] and

---

[5]Llama-7b (fp16) requires 24GB GPU memory for 2048 context tokens and 44GB for 4096 during inference (measured without optimization like flash attention).

[6]Prompts in Mu et al. (2023) refer to task instructions before input texts, so they are usually short.

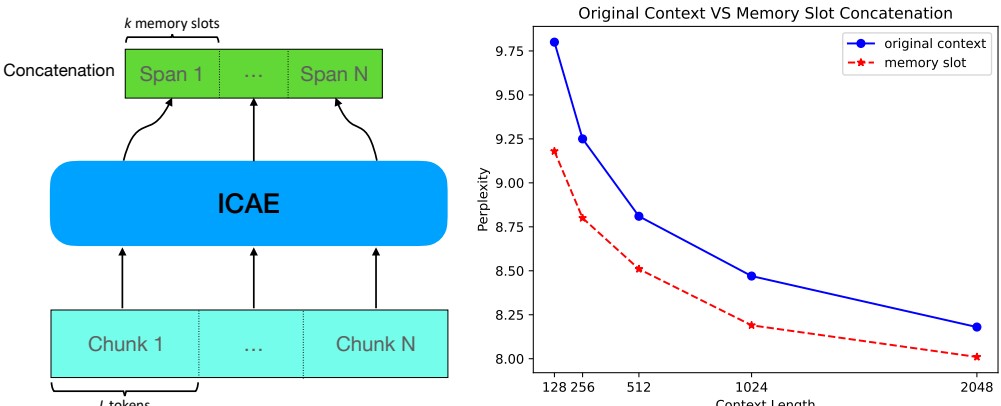

Figure 6: **Left:** Individually compress then concatenate multiple spans of memory slots; **Right:** Perplexity comparison with original contexts and $4\times$ compressed memory slots – for example, 1024-length memory slots are obtained by compressing the original context with a length of 4096 tokens.

thus does not address the real issue of long contexts. Also, this method requires fine-tuning the LLM, and the obtained gist tokens also need to be used within the specially tuned LLM (for gist tokens) and seem not compatible with the untouched LLM. AutoCompressors for recursively compressing long text into summary vectors. Like Mu et al. (2023), the LLM must be tuned to work with generated summary vectors and its training is sophisticated as it involves recursive compression. In contrast, we propose a very simple, straightforward and scalable approach to generating memory slots that can be used in the target LLM with different prompts for various purposes. Moreover, our approach is much more parameter-efficient (i.e., LoRA) for tuning on top of the existing LLM. Additionally, some recent work studies how to compress prompts into more concise natural language (Jiang et al., 2023a), and approaches the context limit with divide-and-conquer methodology (Bertsch et al., 2023; Chen et al., 2023; Song et al., 2024).

Also, there is related work studying compressing indescribable concepts into (vector) tokens for later use in other contexts. Representative work includes Gal et al. (2022) which compresses a vision object into a token and Ge et al. (2023) which compresses a text style into a token.

Considering related work from a boarder perspective of compression, Jiang et al. (2023b) examines $k$NN-based prediction using general-purpose compressors, such as gzip. Delétang et al. (2023) extensively investigates the compression abilities of LLMs, uncovering their potential as versatile predictors, which also provides insights into recent developments in scaling laws and tokenization.

## 5 CONCLUSION AND FUTURE WORK

We propose the In-context Autoencoder (ICAE) to leverage the power of an LLM to highly compress contexts. By generating compact and informative memory slots to represent the original context, the ICAE enables an LLM to acquire more information with the same context length or represent the same content with a shorter context, thereby enhancing the model's capability to handle long contexts as well as reducing computation and memory overheads for inference in many practical scenarios like Retrieval Augmented Generation (Lewis et al., 2020) and advanced prompting methods (Wei et al., 2022; Wang et al., 2023; Zhang et al., 2024). Moreover, ICAE provides insight into how an LLM performs memorization, offering a novel perspective on the connection between the memory of LLMs and humans, and suggesting future research in LLM context management.

Due to computational limitations, our experiments were conducted on Llama models up to 13 billion parameters. As discussed in the paper, ICAE is expected to benefit even more from more powerful LLMs, where it should be able to achieve more significant compression ratios. In the future, we hope to have sufficient computational resources to validate the effectiveness of ICAE on larger and stronger LLMs. In addition, we plan to explore the application of ICAE in multimodal LLMs (as the context length for images, videos, and audio is often much longer and has greater compression potential) with discrete memory slots (which can be either continuous or discrete) for helping unify compact representation across modalities in the era of LLM/AGI.

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

## A  MODEL TRAINING CONFIGURATION

We show how to perform pretraining with the text continuation objective and instruction fine-tuning in Figure 7 and 8.

We train the ICAE on 8 Nvidia A100 GPUs (80GB). The hyperparameters for pretraining and fine-tuning ICAE are presented in Table 8. We by default train the ICAE with bf16.

Table 8: Hyperparameters for training

| Hyperparameter | Value |
|----------------|-------|
| Optimizer | AdamW |
| learning rate | 1e-4 (pretrain); 5e-5 (fine-tuning) |
| batch size | 256 |
| warmup | 300 |
| #updates | 200k (pretrain); 30k (fine-tuning) |
| clip norm | 2.0 |

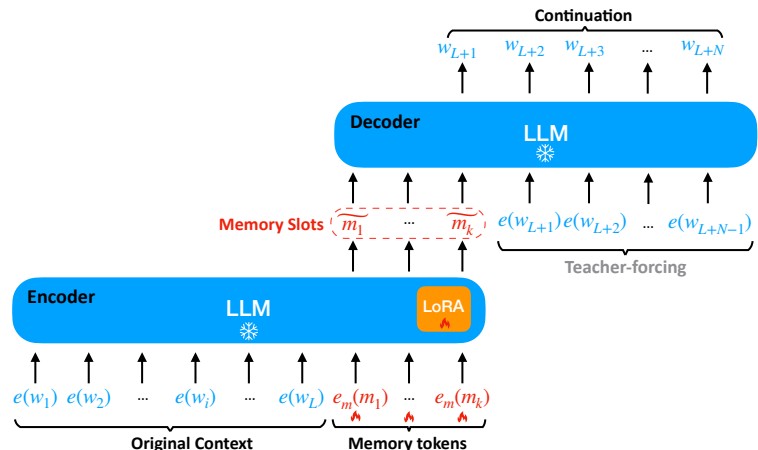

Figure 7: Pretraining with the text continuation objective to predict next tokens

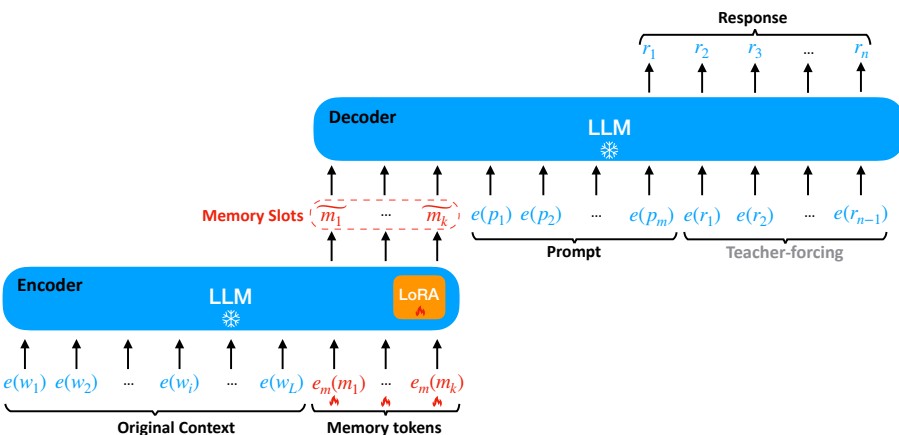

Figure 8: Instruct fine-tuning of the ICAE to make its produced memory slots interact with prompts for accomplishing various purposes in the target LLM. In this figure, $(p_1, \ldots, p_m)$ denotes the prompt tokens and $(r_1, \ldots, r_n)$ denotes the response tokens.

## B   PROFILING SETUP

We test the latency (Section 3.3.2) on 1 Nvidia A100 GPU (80GB). The test machine has the CPU of AMD EPYC™ 7413 with 24 cores and 216GB RAM. The runtime configuration is python=3.9, pytorch=2.0.1, cuda=11.7, cudnn=8.5.

## C   PROMPT-WITH-CONTEXT DATASET

We introduce the PROMPT-WITH-CONTEXT (PwC) dataset where each sample entry is a triple (text, prompt, answer), as depicted in Figure 9. To construct this dataset, we first sample 20k texts from the Pile dataset. Then, for each text, we employ the GPT-4 to provide 15 prompts (10 specific prompts and 5 general prompts) about the text and give the corresponding answers. The prompt instructing the GPT-4 is outlined in Listing 1.

The dataset is composed of 240k examples for training purposes, with an additional 18k examples for testing. The context length distribution of test samples is presented in Table 10.

Listing 1: Prompt used by GPT4 API to generate the PwC dataset.

```
Design 10 prompts specified to the above text to test understanding of
the above text. These prompts should be diverse and cover as many
```

**Context**

*As artificial intelligence becomes an increasingly powerful force, some of the world's biggest companies are worrying about how the technology will be used ethically, and how the public will perceive its spread. To combat these problems (among others), five tech companies — Google, Amazon, Microsoft, Facebook, and IBM — set up a research group called the Partnership on AI.*

*…*

*New trustees joining include Dario Amodei of OpenAI — another industry AI research group founded by Elon Musk and Y Combinator president Sam Altman — and Carol Rose of the ACLU.*

*…*

*AI taking white collar jobs, eroding trust in public media, becoming embedded in public institutions like the courts and hospitals: these are the sorts of problems facing the industry in the future.*

**Prompt:** List the five tech companies that initially set up the Partnership on AI.

**Answer:** *Google, Amazon, Microsoft, Facebook and IBM.*

**Prompt:** What are potential challenges the AI industry might face in the future?
**Answer:** *Challenges include AI taking white collar jobs, eroding trust in public media, and becoming embedded in public institutions like courts and hospitals..*

**Prompt:** Which organization did Dario Amodei previously work for?

**Answer:** *OpenAI*

Figure 9: Construction of the PwC dataset: we use the GPT-4 to generate a variety of prompt-answer pairs according to contexts. The resulting dataset is used for instruction fine-tuning (240k for training) and evaluation (18k for testing) in this work.

```
aspects (e.g., topic, genre, structure, style, polarity, key information
and details) of the text as possible. The first half of these prompts
should be like an instruction, the other should be like a question. In
addition to the prompts specified to the above text, please also design
5 general prompts like "rephrase the above text", "summarize the above
text", "write a title for the above text", "extract a few keywords for
the above text" and "write a paragraph (i.e., continuation) that follows
the above text". Each prompt should be outputted in the following
format: [{"prompt": your generated prompt, "answer": the answer to the
prompt}]
```

## D  GPT-4 EVALUATION

According to Mu et al. (2023), we formulate an evaluation prompt to be used with the GPT-4 API. The prompt, as illustrated in Listing 2, consists of a task description along with three specific examples. We supply GPT-4 with a text, a prompt, and two distinct model-generated responses. The task for GPT-4 is to determine the superior answer or recognize a tie. The chosen examples encompass scenarios where Assistant A performs better, Assistant B performs better, and when a tie occurs. This methodology enables us to effectively assess[7] the model's quality. Specially, the orders where the model responses are presented to the GPT-4 are swapped randomly to alleviate bias, as Touvron et al. (2023b) did.

Listing 2: Prompt for the GPT-4 evaluation. This prompt consists of a description of the task and three specific examples.

```
Given a piece of text, an instruction for this text, and two AI
assistant answers, your task is to choose the better answer and provide
reasons. Evaluate the answers holistically, paying special attention to
```

---

[7]We find the GPT-4 rater tends to prefer longer responses, aligning with observations from recent work such as Zhao et al. (2024). Given that ICAE's responses are generally short (due to instruction fine-tuning with the PwC dataset), its actual performance should be better than the numbers reported in the evaluation.

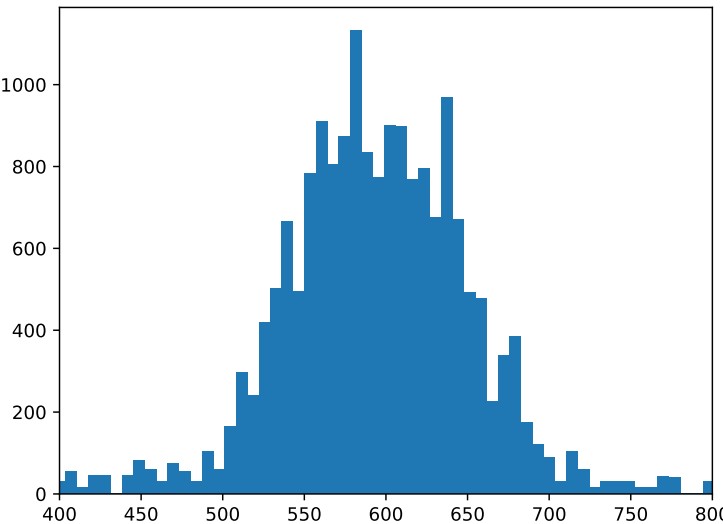

Figure 10: The context length distribution of test samples: Most samples are longer than 500 tokens.

```
whether the response (1) follows the given instruction and (2) is
correct. If both answers correctly respond to the prompt, you should
judge it as a tie.

Example 1:
```
Text: We report the development of GPT-4, a large-scale, multimodal
model which can accept image and text inputs and produce text outputs.
While less capable than humans in many real-world scenarios, GPT-4
exhibits human-level performance on various professional and academic
benchmarks, including passing a simulated bar exam with a score around
the top 10% of test takers. GPT-4 is a Transformerbased model
pre-trained to predict the next token in a document. The post-training
alignment process results in improved performance on measures of
factuality and adherence to desired behavior. A core component of this
project was developing infrastructure and optimization methods that
behave predictably across a wide range of scales. This allowed us to
accurately predict some aspects of GPT-4's performance based on models
trained with no more than 1/1,000th the compute of GPT-4.
Prompt: What is GPT4?
Assistant A: GPT4 is a large-scale language-trained transformer-based
model.
Assistant B: GPT4 can produce outputs.
```

Your output should be:
```
{"reason": "The instruction asks what GPT4 is, and from the original
text, we know that GPT4 is a multimodal, large-scale model that can
generate text. Therefore, Assistant A is the closer answer, while
Assistant B did not follow the instruction well in providing a
response.", "choice": "A"}
```

Example 2:
```
Text: Making language models bigger does not inherently make them better
at following a user's intent. For example, large language models can
generate outputs that are untruthful, toxic, or simply not helpful to
the user. In other words, these models are not aligned with their users.
```

In this paper, we show an avenue for aligning language models with user intent on a wide range of tasks by fine-tuning with human feedback. Starting with a set of labeler-written prompts and prompts submitted through the OpenAI API, we collect a dataset of labeler demonstrations of the desired model behavior, which we use to fine-tune GPT-3 using supervised learning. We then collect a dataset of rankings of model outputs, which we use to further fine-tune this supervised model using reinforcement learning from human feedback. We call the resulting models InstructGPT. In human evaluations on our prompt distribution, outputs from the 1.3B parameter InstructGPT model are preferred to outputs from the 175B GPT-3, despite having 100x fewer parameters. Moreover, InstructGPT models show improvements in truthfulness and reductions in toxic output generation while having minimal performance regressions on public NLP datasets. Even though InstructGPT still makes simple mistakes, our results show that fine-tuning with human feedback is a promising direction for aligning language models with human intent.
Prompt: Write a title for the above text.
Assistant A: Improving Fine-Tuning for Language Models: A GPT-3-inspired Approach
Assistant B: Training language models to follow instructions with human feedback
```

Your output should be:
```
{"reason": "This text discusses how to make large language models follow user instructions better, and Assistant B's response is more in line with the meaning of the text.", "choice": "B"}
```

Example 3:
```
Text: Though technology companies announced massive layoffs last year, 2023 has been much worse. Layoffs have far outpaced last year's cuts, as tech giants including Amazon, Facebook parent company Meta, Microsoft, Google, IBM, SAP, and Salesforce – as well as many smaller companies – announce sweeping job cuts. The problem: Big Tech went on a hiring binge during the pandemic when lockdowns sparked a tech buying spree to support remote work and an uptick in e-commerce, and now they face revenue declines. Although global IT spending is forecast to rise in 2023, with enterprise software and IT services experiencing the greatest growth, the overall increase is expected to be modest, with data center systems and communications services growing by less than 1%, according to market research firm Gartner. Meanwhile hardware sales are forecast to decline. Continuing supply chain issues, inflation, and the war in Ukraine are also having an impact on both business and consumer spending, leading to fears of recession. According to data compiled by Layoffs.fyi, the online tracker keeping tabs on job losses in the technology sector, 784 tech companies have laid off about 208,181 staff so far this year, compared to 164,411 layoffs last year.
Prompt: How many job losses have occurred in the technology sector so far in 2023?
Assistant A: 208,181 job losses have occurred in the technology sector so far in 2023.
Assistant B: 208,181.
```

Your output should be:
```
{"reason": "Both answers are acceptable and correct. They should be a tie.", "choice": "Tie"}
```
Your response should only be in the JSON format above; THERE SHOULD BE NO OTHER CONTENT INCLUDED IN YOUR RESPONSE. Write the "reason" key before writing the "choice" key, so that you think step-by-step before

```
making your decision. KEEP YOUR REASONING BRIEF. Again, don't favor
either A or B if they are both acceptable and correct -- judge a tie
instead.
```

The prompt that the GPT-4 uses to generate 128-token summary is as follows:

```
"Write a summary for the above text.  Your summary should not
exceed 100 words but should include as much information of the
original text as possible."
```

We show examples of the GPT-4 evaluation on a pretrained and a non-pretrained ICAE in Table 9.

Table 9: Examples of outputs by the target LLM (i.e., Llama) conditioning on memory slots ($k = 128$) produced by the pretrained and non-pretrained ICAE. The highlighted parts are not faithful to the context.

---

**Passage 1 (514 tokens):**
French senior civil servant arrested on suspicion of spying for North Korea

November 27, 2018 by Joseph Fitsanakis

A senior civil servant in the upper house of the French parliament has been arrested on suspicion of spying for North Korea, according to prosecutors. The news of the suspected spy's arrest was first reported on Monday by Quotidien, a daily politics and culture show on the Monaco-based television channel TMC. The show cited "a judicial source in Paris" and said that France's domestic security and counterintelligence agency, the General Directorate for Internal Security (DGSI), was in charge of the espionage case.

**(omitted 233 words)**

The senior civil servant faces up to 30 years in prison if found guilty of espionage.

---

**Prompt:** What is the maximum prison sentence Quennedey could face if found guilty?

---

**Assistant A (pretrained ICAE):** Quennedey could face up to 30 years in prison if found guilty.

---

**Assistant B (non-pretrained ICAE):** Quennedey could face up to three years in prison if found guilty.

---

**Answer (by the GPT-4):** Up to 30 years.

---

**GPT-4 evaluation:** Assistant A correctly states the maximum prison sentence from the text, while Assistant B provides an incorrect number.

---

**Passage 2 (506 tokens):**
With Red Hat, IBM to become the leading hybrid cloud provider Watch Now

After IBM acquired Red Hat, I suggested IBM paid $34 billion for the Linux power so it could become a hybrid-cloud power. With the news that Red Hat will acquire NooBaa, a hybrid-cloud, data-storage company, it's become clearer than ever that the IBM-Red Hat deal is all about the hybrid cloud.

**(omitted 208 words)**

As Ranga Rangachari, Red Hat's vice president of Storage and Hyperconverged Infrastructure, said in a statement:

"Data portability is a key imperative for organizations building and deploying cloud-native applications across private and multiple clouds. NooBaa's technologies will augment our portfolio and strengthen our ability to meet the needs of developers in today's hybrid and multicloud world. We are thrilled to welcome a technical team of nine to the Red Hat family as we work together to further solidify Red Hat as a leading provider of open hybrid-cloud technologies.

---

**Prompt:** Discuss the role of Ranga Rangachari in the context of the text

---

**Assistant A (pretrained ICAE):** Ranga Rangachari is the vice president of Storage and Hyperconverged Infrastructure at Red Hat. He is responsible for the company's storage and hyperconverged offerings, and he welcomes the nine-member team of technical experts from NoaBaa to join Red Hat as a part of the IBM acquisition.

---

**Assistant B (non-pretrained ICAE):** Ranga Rangachari is the CEO of Red Hat, and he is excited about the acquisition of NooBaa as it will help Red Hat become a leader in hybrid cloud management. He believes that NooBaa's technology will enable Red Hat to provide a single, comprehensive platform for managing data across multiple clouds, and that this will be a significant differentiator for Red Hat in the market.

---

**Answer (by the GPT-4):** Ranga Rangachari is the vice president of Storage and Hyperconverged Infrastructure at Red Hat. He made a statement regarding the acquisition of NooBaa, highlighting the importance of data portability in today's hybrid and multicloud world and how NooBaa's technologies would strengthen Red Hat's ability to meet the needs of developers.

---

**GPT-4 evaluation:** Assistant A correctly identifies Ranga Rangachari's role as the vice president of Storage and Hyperconverged Infrastructure at Red Hat and accurately describes his statement about the acquisition of NooBaa. Assistant B incorrectly states that Ranga Rangachari is the CEO of Red Hat.

---

