# OpenReview forum: "In-context Autoencoder for Context Compression in a Large Language Model"
_ICLR.cc/2024/Conference — ICLR 2024 poster_

### Official Review · Reviewer_DLaD · 2023-10-26

**Soundness:** 3 good
**Presentation:** 3 good
**Contribution:** 2 fair
**Rating:** 6
**Confidence:** 4

**Summary:**

This paper enables LLM to handle long contexts by compressing the context into shorter memories. To achieve this, another encoder LLM (in addition to the original decoder LLM) is fine-tuned. The memories compressed by the encoder LLM are used to reconstruct the original context and generate future sentences.

**Strengths:**

- The paper tackles a timely and important problem of LLM.
- The proposed method is simple and intuitive.
- The paper is well-written in general.

**Weaknesses:**

**Comparison with summarization**

The paper compresses the contexts into black-box memories. Instead, one can simply apply a summarization model (e.g., GPT-4) to summarize the context and feed it to the model. While Figure 2 claims that black-box memory can be more compressive than words, this claim is not supported by the experiments. Besides, summarization has a merit in that it can be applied to black-box API models, unlike this method, which is only applicable to open-source white-box models.

---
**Still need quadratic complexity**

The model still needs to forward the entire context to compress the memories, which requires quadratic complexity over the context length. The paper partly addresses this issue using multiple spans of memories in Figure 6, but the results are preliminary. Presenting this divide-and-conquer approach as the primary method and providing a deeper analysis would extend the method to be applied for extremely long contexts. Additionally, it's worth noting that there are several concurrent submissions considering the divide-and-conquer concept [1-3].

[1] Unlimiformer: Long-Range Transformers with Unlimited Length Input\
[2] Walking Down the Memory Maze: Beyond Context Limit through Interactive Reading\
[3] Hierarchical Context Merging: Better Long Context Understanding for Pre-trained LLMs

---
**Ohter long context works**

Though concurrent, discussing other techniques for long context LLM in the related work section would help readers grasp the positioning of this paper. I can see many works on OpenReview and arXiv, such as [4-5], to name a few.

[4] LongLoRA: Efficient Fine-tuning of Long-Context Large Language Models\
[5] Functional Interpolation for Relative Positions improves Long Context Transformers

---
**Editorial comments**

- Using a different color for linkcolor would help readers better recognizing the link to the tables.
- Putting tables/figures at top instead of here would help readers to better skim the experiments.

**Questions:**

My main concerns/questions are:
1. Additional validation of the multiple-span approach. Especially, checking the generated examples and practical benchmarks, such as QA, instead of relying solely on perplexity. I wonder if this multiple-span approach would work really well, even though the model is fine-tuned with only a single span.
2. Comparison with the summarization baseline, which simply compresses the context into words instead of black-box memories.

---

> ### Author Response · Authors · 2023-11-20
> **Response to Reviewer DLaD**
>
> We greatly appreciate your recognition and praise for this work, as well as the constructive suggestions you have provided to help us improve it. We hope our following responses can clarify some details and address your concerns, thereby improving your evaluation of this work.
>
> > Weakness 1: The paper compresses the contexts into black-box memories. Instead, one can simply apply a summarization model (e.g., GPT-4) to summarize the context and feed it to the model. While Figure 2 claims that black-box memory can be more compressive than words, this claim is not supported by the experiments. Besides, summarization has a merit in that it can be applied to black-box API models, unlike this method, which is only applicable to open-source white-box models.
> >
>
> We have experimental results to show that black-box memories can be more compressive than a summary. Firstly, both Figure 4 and 5 demonstrate that using 128-length black-box memories can almost losslessly restore the original 512-length context (with BLEU>99%), which cannot be achieved using a 128-length summary. Secondly, in the last row of Table 6, we compared 128 black-box memory and 128-token summary, which also confirmed that black-box memory is more compact and informative than a summary with the equal length.
>
> > Weakness 2: Still need quadratic complexity
> >
>
> Our motivation is not to improve quadratic complexity, but to effectively reduce the context length to make the context more compact and informative. You are correct that a divide-and-conquer strategy can make this method applicable to super-long contexts. We appreciate the references you recommended and will extend our discussion on this in the revised version to make this paper's position and contribution more clear.
>
> > Weakness 3: Other long context works
> >
>
> Thank you for your reminder and recommendations. We will include and extend the discussion of these contemporary papers in the revised version.
>
> > Weakness 4: Editorial comments
> >
>
> We appreciate your suggestions on the presentation of this paper. We will fix these issues in the next version according to your advice.
>
> > Q1: Additional validation of the multiple-span approach. Especially, checking the generated examples and practical benchmarks, such as QA, instead of relying solely on perplexity. I wonder if this multiple-span approach would work really well, even though the model is fine-tuned with only a single span.
> >
>
> Thank you for your suggestion. As we respond to Weakness 2, we will extend the discussion on the multiple span approach.
>
> > Q2: Comparison with the summarization baseline, which simply compresses the context into words instead of black-box memories.
> >
>
> As we respond to Weakness 1, we have the experiment results in Figure 4/5 and Table 6 to demonstrate that the memory slots are more compact and informative than the summary with the equal length.

---

> > ### Comment · Reviewer_DLaD · 2023-11-22
> > **Response to the Rebuttal**
> >
> > Thank you for your response. I carefully read the other reviews and the rebuttal. I still believe that the technical novelty of the work could be improved by delving deeper into the multi-span approach. Nevertheless, the proposed simple method appears to be effective and is soundly validated through experiments. For this reason, I have raised my score to 6.

---

> > > ### Author Response · Authors · 2023-11-22
> > >
> > > We really appreciate your acknowledgment of our response and positive feedback to this work. We will follow your suggestion to improve this work with more in-depth investigation on the multi-span setting.

---

### Official Review · Reviewer_kMU9 · 2023-10-28

**Soundness:** 3 good
**Presentation:** 3 good
**Contribution:** 3 good
**Rating:** 5
**Confidence:** 4

**Summary:**

In this paper, the authors propose a new approach for compressing the context of LLMs:

* The architecture involves an encoder, based on existing LLMs, and a decoder which is also a LLM. The encoder takes in the context, and a set of special memory tokens. The output embeddings of the memory tokens are used as the compressed representation of the context. This representation is passed to the decoder before the target token embeddings.

* The paper proposes three training schemes: the autoencoding pretraining, text continuation pretraining and instruction fine-tuning. The autoencoding pretraining trains the decoder to reconstruct the encoder inputs, with only the compressed memory as the conditioning. The text continuation involves continuing the context passed to the encoder. Finally the model is fine-tuned for instruction following using the Prompt-with-Context dataset also proposed in the paper.

* Finally there are series of analyses on how compression ratio affects the performance, and memory encoding behavior of this autoencoder.

**Strengths:**

The problem, namely how to compress context and form memory for LLMs, is an important topic, and the solution proposed in this paper is sound. This exact formulation is novel, as well as the training techniques and the dataset. The analysis on how compression ratio affects the performance on the reconstruction task, and the analysis on how different text format could affect memorization is also very insightful. Finally, the paper is generally easy to follow, with important information provided in both the main text and appendix.

**Weaknesses:**

I found the baseline somewhat lacking. There are a few other papers targeting the exact or adjacent problems, such as Gisting, Recurrent memory transformer, compressive transformer, auto compressor. However, the paper did not compare with any of them, making it hard to judge the tradeoffs between different methods. It'll be nice to at least compare to one of the existing methods, such as gisting, especially as it claims to achieve up to 26x compression. Further, it'll be nice to have ablation studies on various design choices, such as embedding the input tokens into the same space as the decoder for the encoder. More details on weakness will be in the "Questions" section. I'm willing to increase the score if the aforementioned weaknesses are addressed.

**Questions:**

1. I likely have missed something in the paper. It's unclear to me whether the authors initialized the encoder LLM from the same checkpoint as the decoder LLM? If that's the case, how did the author address the difference in autoregressive attention in the decoder (and the base LLM, such as LLAMA)  and bidirectional attention potentially used for the encoder.
2. Why not use an encoder-decoder language model such as T5 for this?
3. As mentioned above, does it matter for the encoder's input embeddings to be in the same space as the decoder's input embeddings?
4. What's the actual size of each of the pretraining tasks, in terms of number of examples or number of tokens consumed.
5. I would imagine the autoencoding task to be relatively easy to "learn". How many steps or number of examples does it take for the model to converge?
6. Are the two pretraining tasks interleaved or done in a sequential order?
7. For the instruction fine-tuning, have you tried passing the prompt to the encoder instead of the decoder, and what's the effect?
8. Do you initialize the encoder from the same LLM checkpoint as the decoder?
9. Could you provide some intuition on why the loss in the rightmost image in Figure 4 shows a U shaped trend.
10. Could you also provide some explanation on why the first row of table 1 shows a PPLX increase even though there's no compression going on.
11. Regarding the claim that pretrained ICAE suffers less from hallucination than non-pretrained model, I would like to clarify that both variants are trained on the same instruction fine-tuning set right?
12. Have you tried full model fine-tuning? In other words, what's the impact of lora on the final quality?

---

> ### Author Response · Authors · 2023-11-20
> **Response to Reviewer kMU9 (Part 1)**
>
> We greatly appreciate your recognition and constructive suggestions for this work. We hope our following responses can clarify some details and address your concerns, thereby improving your evaluation of this work.
>
> > Weakness: I found the baseline somewhat lacking. There are a few other papers targeting the exact or adjacent problems, such as Gisting, Recurrent memory transformer, compressive transformer, auto compressor.
>
> As you can see in our references, we have properly cited these papers. The Recurrent Memory Transformer (RMT) and Compressive Transformer primarily focus on exploring a new network architecture or learning method to address the long context issue, as introduced in Section 1. Their methods, however, are not readily applicable to compress long text for an existing Language Model (LLM) such as Llama-2. This is the fundamental difference between these papers and our work -- our motivation and focus is to compress long context for an existing LLM to use the compressed context to replace the original context during inference.
>
> As for Gisting and Autocompressor, although they also study context compression, they do not compress context for an existing LLM in the same way we do:
>
> Gisting's compression is limited to transform a task description into a few compressed (task) vectors (i.e., gist tokens). It does not involve content compression. Its 26x compression is regarding compressing task description prompt which is usually not very long, as we discussed in Section 5. Moreover, Gisting itself requires fine-tuning the LLM to compress, and its produced gist tokens is not intended for the original LLM but for its fine-tuned LLM, which is not what we aim for.
>
> As for Autocompressor, it does not compress context for the original LLM either. Its motivation and focus are more akin to the Compressive Transformer.
>
> In contrast, our work is designed to compress a long context for an existing LLM. The LLM can directly use the compressed memory slots as the (conditional) context to perform various tasks or goals. This is a more practical scenario **as we don't need to modify anything on a hosted LLM. We only need to pass the memory slots to the LLM in the same way as word embeddings**. **Clearly, none of these four previous papers can be directly applied for this goal, which is the reason why we do not compared with them in this scenario.**
>
> > Q1.  I likely have missed something in the paper. It's unclear to me whether the authors initialized the encoder LLM from the same checkpoint as the decoder LLM? If that's the case, how did the author address the difference in autoregressive attention in the decoder (and the base LLM, such as LLAMA) and bidirectional attention potentially used for the encoder.
> >
>
> The encoder in our paper refers to the encoder in the autoencoder terminology, which is used for encoding text into memory slots. It is not the same as the encoder in the encoder-decoder architecture.
>
> As illustrated in Figure 3, the encoder of ICAE is a frozen LLM (specifically Llama in this paper) + a learnable LoRA + a learnable embedding for memory tokens. The decoder of ICAE is the frozen LLM.
>
> Therefore, the encoder does not use bidirectional attention; instead, it uses causal attention, as in the decoder.
>
> We apologize for the confusion. We'll make it clearer in the revised version.
>
> > Q2.  Why not use an encoder-decoder language model such as T5 for this?
> >
>
> As we mentioned in our response to Q1, our encoder and decoder refer to the encoding and decoding modules of the autoencoder (i.e., ICAE), which are totally different from the encoder/decoder in the encoder-decoder architecture. Our goal is to compress contexts for an existing Large Language Model (LLM), specifically Llama in our experiments. Building the ICAE on the same LLM (Llama) allows us to better align with the LLM to generate a more compressed context.
>
> > Q3.  As mentioned above, does it matter for the encoder's input embeddings to be in the same space as the decoder's input embeddings?
> >
>
> As we previously mentioned in our response to Q1, the encoder's input embeddings and the decoder's input embeddings are indeed in the same space.
>
> > Q4. What's the actual size of each of the pretraining tasks, in terms of number of examples or number of tokens consumed.
> >
>
> We have included these details in Appendix A. For pretraining, our batch size and training steps are listed in Table 9. The number of training tokens can be calculated as 256 * 512 * 200k = 26.2 billion tokens.
>
> > Q5. I would imagine the autoencoding task to be relatively easy to "learn". How many steps or number of examples does it take for the model to converge?
> >
>
> You’re exactly right. Autoencoding is easier to learn than Language Modeling. However, our setting is 4x compression (e.g., using 128 tokens to restore 512 tokens). According to our observations, the validation loss still keeps decreasing even after 150k training steps.

---

> ### Author Response · Authors · 2023-11-20
> **Response to Reviewer kMU9 (Part 2)**
>
> > Q6. Are the two pretraining tasks interleaved or done in a sequential order?
> >
>
> They are interleaved. Sequential training is not good in our setting due to the catastrophic forgetting issue.
>
> > Q7. For the instruction fine-tuning, have you tried passing the prompt to the encoder instead of the decoder, and what's the effect?
> >
>
> Yes, we tried. As long as we mixed such data in our instruction fine-tuning, the ICAE could still perform well because compressing task description prompt is not hard.
>
> > Q8. Do you initialize the encoder from the same LLM checkpoint as the decoder?
> >
>
> Yes, they are the same — they are both Llama. The only difference is that the ICAE encoder adds a learnable LoRA module and a learnable embedding for memory tokens (see Figure 3).
>
> > Q9. Could you provide some intuition on why the loss in the rightmost image in Figure 4 shows a U shaped trend.
> >
>
> Very good question. We internally discussed this observation and we think there are two reasons. First, our pretraining samples are usually long (~500 tokens). Short sequences like 100, 200 are less common in the training data, leading to higher loss. Second, the loss tends to be higher for earlier context — for example, for the 10th word, there are only 9 words to condition on, so its loss tends to be higher. But for the 300th word, there are 299 words to condition on (i.e., richer context), making its prediction easier (i.e., lower loss)
>
> > Q10. Could you also provide some explanation on why the first row of table 1 shows a PPLX increase even though there's no compression going on.
> >
>
> This is because we replaced the original natural language tokens with compressed memory slots, which inevitably causes some loss (even without compression). This is understandable as our memory slots are pretrained on 26.2 billion tokens, while natural language tokens are pretrained on trillions of tokens by Llama.
>
> > Q11. Regarding the claim that pretrained ICAE suffers less from hallucination than non-pretrained model, I would like to clarify that both variants are trained on the same instruction fine-tuning set right?
> >
>
> Yes, the only difference is that one is pretrained with AE+LM before instruction fine-tuning, while the other is not pretrained and is directly trained on the instruction fine-tuning data.
>
> > Q12. Have you tried full model fine-tuning? In other words, what's the impact of lora on the final quality?
> >
>
> We have tried it on a small scale, but due to computational resource limitations, we have not done it on a large scale. However, under the same training steps (10k), we found that the results of full model fine-tuning are better than LoRA. But the advantage of LoRA is that it only adds 1% extra memory when used, which is important for practical applications.

---

> > ### Author Response · Authors · 2023-11-22
> >
> > Dear Reviewer kMU9,
> >
> > We would like to express our sincere gratitude for your detailed feedback and your willingness to consider increasing your rating.
> >
> > We have provided detailed responses to each point you raised, and we hope that these clarifications have dispelled any misunderstandings and align with your expectations. Could you kindly confirm if our rebuttal has adequately addressed your concerns? Should there be any remaining questions or additional feedback, please know that we are fully willing to continue this conversation and provide further clarifications as needed.
> >
> > Thank you for your time and consideration.

---

### Official Review · Reviewer_fbEw · 2023-10-29

**Soundness:** 4 excellent
**Presentation:** 4 excellent
**Contribution:** 3 good
**Rating:** 8
**Confidence:** 5

**Summary:**

This work proposes a to generate compact representation using memory slots to represent the original context. The proposed approach
is ICAE also discusses an auto-encoding object (with lora) to pretrain the llm to generate good memory slots. The authors present several interesting experiments poking at the ability of model to do good compression as well as being able to use the compressed vectors.

**Strengths:**

* Well packaged paper that addresses the prompt compression using detailed experiments.
* Originality: I was happy to see the discussion on auto-encoding ability, as well experimentation on what length of memory slots worked vs not. Also, it was interesting to see the Lora approach to generate memory slots. I hadn't seen that in previous work.
* Quality: Very thorough experiments on different approaches
* Clarity: Well presented
* Significance: Per the experiments, it looks like we should be able to cut-down ~1/4 of the prompt by using dense-encodings instead of raw text without much degradation in performance. This is a pretty good first step to see if we reduce this further more, e.g. 1/10 or so (perhaps with a larger model).

**Weaknesses:**

* I wish the authors were more forthcoming on how this builds on top of existing work. For e.g. the authors only gloss over Chevalier et al. (2023) in passing which covers what the authors proposing to a large extent. Nevertheless, the biggest difference in this work seems to be the thorough experiments which was missing in the other work.

* I encourage the authors to think more on how these memory-slots/dense-encodings can be freely interspersed with raw text. For e.g. it would be a step function improvement if one could write a prompt like "use these docs [m1], [m2].. [mk] to answer the following questions [m1'], [m2'] [m3'].. [mk']"

**Questions:**

1. In table 5, in rows 2, 3, 4 and 5 is column#2 (i.e. "system 2") also fine-tuned on the pwc dataset? I presume so, but I do want to check. Without this, it would be a very slightly unfair comparison if "system 1" got to see the pwc-train set but "system 2" did not.

---

> ### Author Response · Authors · 2023-11-20
> **Response to Reviewer fbEw**
>
> Thank you very much for your appreciation and recognition of this work. As you rightly pointed out, this is our first step towards long context compression, and we are actively working on more ambitious compression — just as you suggest — 10x or even more. We also greatly appreciate your pointing out some issues of this work, and we are very happy to address your concerns to further improve this work:
>
> > I wish the authors were more forthcoming on how this builds on top of existing work
> >
>
> We appreciate your comment and suggestions on the related work discussion, we will extend the discussion about previous work and elaborate on the differences to clarify our contribution.
>
> > I encourage the authors to think more on how these memory-slots/dense-encodings can be freely interspersed with raw text.
> >
>
> This is exactly our ongoing effort. By mixing data in various formats and combinations, we have initially confirmed that this method can be extended to be freely interspersed. We are also looking for an efficient method to generate diverse training data to allow memory slots to be flexibly interacted with natural language in high robustness.
>
> > In table 5, in rows 2, 3, 4 and 5 is column#2 (i.e. "system 2") also fine-tuned on the pwc dataset? I presume so, but I do want to check. Without this, it would be a very slightly unfair comparison if "system 1" got to see the pwc-train set but "system 2" did not.
> >
>
> All the llama-2-chat models in the column of system 2 are fine-tuned with the PwC dataset using LoRA.

---

### Official Review · Reviewer_anmc · 2023-11-02

**Soundness:** 4 excellent
**Presentation:** 3 good
**Contribution:** 4 excellent
**Rating:** 8
**Confidence:** 4

**Summary:**

This paper proposes the in-context autoencoder (ICAE) to compress a long context into memory slots and use them for shorter prompts. ICAE requires only a small number of additional parameters by the LoRA approach with a fixed LLM. ICAE is first pre-trained with autoencoding and language modeling objectives on unsupervised text corpus, and then instruction fine-tuned with the Prompt-with-Context (PwC) dataset, newly introduced in the paper.

**Strengths:**

The idea of compression with autoencoding is intuitive and well combined with existing LLMs. The assumption that better LLMs would allow more compact context compression is quite interesting in connection with human memorization and validated in the paper. ICAE can be incorporated into any LLMs and has diverse potential applications, including long context modeling and CoT reasoning.

**Weaknesses:**

One drawback is that ICAE should be anyway pre-trained, despite few additional parameters and fewer training steps than LLM pre-trainings. Compression will be extremely beneficial if we use the same context multiple times. However, otherwise, it might only incur overhead.

**Questions:**

What is the format of memory slots, and how are they generated from the model? I could not find where it is described, but they should be vectors on the embedding space to be fed into the decoder as input. Are they representations before the final softmax layer to get the next word probability distribution? It might be meaningless, but what will we see if we map each memory slot vector to the most likely word?

If I understand correctly, since memory tokens are fixed for any context input, memory slots can be generated in parallel.  What memory tokens e_m have learned?
Is it correct that memory slots are order-dependent because they use the same causal attention of language models? If yes, is it optimal because we have access to all information when we create memory slots?

Based on the fact that the amount of information varies depending on the content of the context, not solely depending on its length, is it possible to have an adaptive size of memory slots for more compact compression?

I agree that AE and LM objectives are straightforward for pre-training ICAE. I wonder if you have tried different objectives. Assuming we can pre-train ICAE from scratch instead of further pre-training from a pre-trained LLM, what will be the best way? Still the same objectives?

In Section 3.3.1, you mentioned the scalability in terms of context length. Are models trained separately for the sequence lengths? How does ICAE generalize well to different (L, k) combinations that are different from the one used for training?

You mentioned that the maximal token length is 512 in the Data paragraph in Section 3.1.  However, most samples are longer than 512 tokens, according to Figure 10. Did you truncate them by 512?

In Table 5, do you have the result of System 1 - Llama2-7b-chat (without ICAE) and System 2 - GPT-4 for reference?

What value of N did you use for the text continuation training?

Training requires 8 A100 GPUs with 80GB. Can you describe the approximate training cost (time and money)?

(minor) the first paragraph in Section 3.3.3: Table 6(Right) -> Figure 6 (Right)

---

> ### Author Response · Authors · 2023-11-20
> **Response to Reviewer anmc (Part 1)**
>
> We greatly appreciate your recognition and constructive feedback on our work. We hope our responses below can address your concerns and answer your questions.
>
> > One drawback is that ICAE should be anyway pre-trained, despite few additional parameters and fewer training steps than LLM pre-trainings. Compression will be extremely beneficial if we use the same context multiple times. However, otherwise, it might only incur overhead.
> >
>
> When we use the same context (i.e., cached in advance), we can significantly save on latency/memory overhead (over 7x speedup, as we discussed in Section 3.3.2). This scenario is **common and practical**: For example, a customized ChatGPT for a specific user can compress the user's chat history in advance and use the compressed memory during inference to maintain the long-term memory without online latency overhead for compression. Regarding your concern about the need to recompress for different contexts, different contexts indeed require an additional compression step. However, **as we show in Table 8, this additional compression still brings significant end-to-end speedup compared to using the original context when generating a long sequence**, making it still a highly practical method.
>
> > What is the format of memory slots, and how are they generated from the model? I could not find where it is described, but they should be vectors on the embedding space to be fed into the decoder as input. Are they representations before the final softmax layer to get the next word probability distribution? It might be meaningless, but what will we see if we map each memory slot vector to the most likely word?
> >
>
> Memory slots are indeed vectors in the embedding space that are fed into the decoder as input. They are the final output of the ICAE encoder (before softmax) in the positions of memory tokens. When we map the memory slot vector to the vocabulary space, we find that memory slots tend to retain some keyword information from the original context and are relatively concentrated in the front part of the original context. This can be explained by the memorization intuition we discussed in our paper, where the model will only remember keywords that provide crucial hints.
>
> > If I understand correctly, since memory tokens are fixed for any context input, memory slots can be generated in parallel. What memory tokens e_m have learned? Is it correct that memory slots are order-dependent because they use the same causal attention of language models? If yes, is it optimal because we have access to all information when we create memory slots?
> >
>
> Yes, you’re exactly correct. They are generated in parallel. They are order-dependent and use causal attention. We think it is an appropriate way to create memory slots just as people memorize in their working memory.
>
> > Based on the fact that the amount of information varies depending on the content of the context, not solely depending on its length, is it possible to have an adaptive size of memory slots for more compact compression?
> >
>
> As shown in Table 3, texts with lower PPL have greater compression potential. This is because lower PPL mean higher certainty and less difficulty to memorize. Your suggestion is very correct. We are working on adaptive memory slots to dynamically adjust the compression rate according to the difficulty of the text (PPL), in order to achieve the best compression performance without affecting the results.

---

> ### Author Response · Authors · 2023-11-20
> **Response to Reviewer anmc (Part 2)**
>
> > Pretraining ICAE from scractch?
> >
>
> We have tried pretraining an ICAE with the same objective using a GPT-style model from scratch with a similar number of parameters as we used in this paper. However, its results are weaker than the ICAE based on a pretrained LLM. Although we have not found a more efficient and scalable objective than AE/LM, we do not deny that there may be a better training objective to pretrain ICAE, as you suggested.
>
> > In Section 3.3.1, you mentioned the scalability in terms of context length. Are models trained separately for the sequence lengths? How does ICAE generalize well to different (L, k) combinations that are different from the one used for training?
> >
>
> How well ICAE generalizes to different (L, k) combinations depends on how we train the ICAE. If we only use L→k to train the ICAE, it will not generalize well to different length combinations. However, if we mix different lengths during the pretraining phase, the model can generalize well to different lengths.
>
> > You mentioned that the maximal token length is 512 in the Data paragraph in Section 3.1. However, most samples are longer than 512 tokens, according to Figure 10. Did you truncate them by 512?
> >
>
> Yes, in the 512 fixed-length experiment group, we truncate to 512 and then compare the results fairly. As most samples are longer than 512 tokens, it is reliable to say ICAE is performing a 4x compression for the majority of test cases.
>
> > In Table 5, do you have the result of System 1 - Llama2-7b-chat (without ICAE) and System 2 - GPT-4 for reference?
> >
>
> We do have the result of System 1 - Llama2-7b-chat (without ICAE) and System 2 - GPT-4 for reference as follows. We’ll include this result in the next version.
>
> |     System 1    | System 2 |  Win | Lose |  Tie |
> |:---------------:|:--------:|:----:|:----:|:----:|
> | Llama-2-7b-chat |   GPT-4  | 19.3 | 24.5 | 56.2 |
>
> > What value of N did you use for the text continuation training?
> >
>
> Do you mean Figure 6(Left)? In our experiments for text continuation training, N ranges from 1 to 4.
>
> > Training requires 8 A100 GPUs with 80GB. Can you describe the approximate training cost (time and money)?
> >
>
> Training does not require 8 A100 GPUs with 80GB, it can be done with just 1 A100 (because we use LoRA, there is no need to control the GPU memory with parallelism techniques). We used 8 A100 GPUs just for speeding up the pretraining process, which takes about 4 days on 26.2 billion tokens.
>
> > (minor) the first paragraph in Section 3.3.3: Table 6(Right) -> Figure 6 (Right)
> >
>
> Thank you very much for pointing out the error in Section 3.3.3. We will correct it in the next version.

---

### Meta-Review · Area_Chair_k4xr · 2023-12-10

**Metareview:**

The submission introduces an auto encoder for LLMs that can efficiently compress long sequence of tokens into a small number of black box memory tokens. The method is trained in a novel but reasonable way by concatenating memory tokens to the end of a sequence before running an encoder over it, then reconstructing the sequence with a decoder that can access only the memory. A particularly nice property is that it can be applied to existing pre-trained LLMs (constructing the encoder by adding LoRA to the LLM) , rather than needing architectural modifications and re-training from scratch. The main point that could be improved is more discussion/positioning of how this paper relates to numerous similar papers (though I agree that this work is novel).

**Justification For Why Not Higher Score:**

I don't mind if this is bumped up to a spotlight

**Justification For Why Not Lower Score:**

This is a clear accept

---

### Decision · Program_Chairs · 2024-01-16

Accept (poster)